# Prolonged intracellular accumulation of light-inducible nanoparticles in leukemia cells allows their remote activation

Carlos Boto[1,2,*], Emanuel Quartin[1,2,*], Yijun Cai[3], Alberto Martín-Lorenzo[4,5], María Begoña García Cenador[5,6], Sandra Pinto[1,2], Rajeev Gupta[7], Tariq Enver[7], Isidro Sánchez-García[4,5], Dengli Hong[3], Ricardo Pires das Neves[1,2] & Lino Ferreira[1,8]

Leukaemia cells that are resistant to conventional therapies are thought to reside in protective niches. Here, we describe light-inducible polymeric retinoic acid (RA)-containing nano-particles (NPs) with the capacity to accumulate in the cytoplasm of leukaemia cells for several days and release their RA payloads within a few minutes upon exposure to blue/UV light. Compared to NPs that are not activated by light exposure, these NPs more efficiently reduce the clonogenicity of bone marrow cancer cells from patients with acute myeloid leukaemia (AML) and induce the differentiation of RA-low sensitive leukaemia cells. Importantly, we show that leukaemia cells transfected with light-inducible NPs containing RA can engraft into bone marrow *in vivo* in the proximity of other leukaemic cells, differentiate upon exposure to blue light and release paracrine factors that modulate nearby cells. The NPs described here offer a promising strategy for controlling distant cell populations and remotely modulating leukaemic niches.

[1] CNC-Center for Neurosciences and Cell Biology, University of Coimbra, 3004-517 Coimbra, Portugal. [2] 3Is—Institute for Interdisciplinary Research, University of Coimbra, 3030-789 Coimbra, Portugal. [3] Key Laboratory of Cell Differentiation and Apoptosis of National Ministry of Education, Department of Pathophysiology, Shanghai Jiao Tong University School of Medicine, Shanghai 200025, China. [4] Experimental Therapeutics and Translational Oncology Program, Instituto de Biologia Molecular y Celular del Cancer (IBMCC), CSIC/University of Salamanca, Salamanca, Spain. [5] Institute of Biomedical Research of Salamanca (IBSAL), Hospital Virgen de La Vega, 37007 Salamanca, Spain. [6] Department of Surgery, University of Salamanca, 37007 Salamanca, Spain. [7] UCL Cancer Institute, University College London, WC1E 6DD London, UK. [8] Faculty of Medicine, University of Coimbra, 3004-504 Coimbra, Portugal. * These authors contributed equally to this work. Correspondence and requests for materials should be addressed to R.P.d.N. (email: ricardo.neves@biocant.pt) or to L.F. (email: Lino@uc-biotech.pt).

The differentiation of leukaemia cells is a therapeutic strategy often used in the clinic to eradicate blood cancers. The concentration of the agent used for inducing leukaemia cell differentiation and the spatio-temporal control of its application are important variables for the success of this therapeutic approach[1]. Induction of leukaemia cell differentiation by RA is a therapeutic strategy that has been used with great success in the treatment of acute promyelocytic leukaemia (APL)[2,3]. RA activates nuclear RA receptors (RARs) that induce cell growth arrest and differentiation[4]. Despite its clear therapeutic efficacy, approximately 25% of patients receiving RA will develop serious complications, such as 'differentiation syndrome'[5]. Hence, there is a need for more effective formulations to deliver RA into leukaemia cells while preventing RA side effects. In addition, leukaemia cells resistant to conventional therapies reside in microenvironmental niches in the bone marrow that are difficult to access by therapeutic interventions[6]. New strategies are required to address these problems.

Nanoparticles (NPs) that disassemble in response to light[7-9] offer a promising approach for reducing the side effects of conventional therapies and increasing access of therapeutic agents to the target cells. Recently, light-inducible NPs have been reported to target solid tumours due to their specific accumulation in tumour vasculature after intravenous injection[10]. However, such an approach is not applicable to leukaemia. The hypotheses of the present work are: (i) light-inducible NPs containing RA may be a more effective strategy for differentiating leukaemia cells because they release high and more effective concentrations of RA in a short period of time (within minutes) after NP disassembly, and (ii) light-inducible NPs containing RA accumulated in the cytoplasm of leukaemia cells may offer a unique opportunity to remotely differentiate these cells in leukaemic niches in the bone marrow, which in turn may interfere with the differentiation profile of leukaemia cells in a paracrine manner.

Here, we describe light-inducible polymeric NPs containing RA that effectively disassemble within cells after light activation. These NPs accumulate in the cytoplasm of leukaemia cells for more than 6 days. They are internalized primarily through a clathrin-mediated mechanism and at minor extent by macropinocytosis. They escape in few hours the endolysosomal compartment and accumulate in cell cytoplasm. We show that these NPs are more efficient and quicker at inducing transcription from the RARE-luciferase locus than RA in solution. We further show that these NPs can be activated to release RA in vivo in a highly controlled manner. Finally, we demonstrate that leukaemia cells transfected with these cells can home in the bone marrow in the same niche as other leukaemia cells, differentiate after blue laser activation and modulate the activity/phenotype of the resident leukaemia cells.

## Results

**Photo-disassembly and release properties of light-inducible NPs.** To prepare light-inducible polymeric NPs, poly(ethyleneimine) (PEI) was initially derivatized with 4,5-dimethoxy-2-nitrobenzyl chloroformate (DMNC), a light-sensitive photochrome (Fig. 1a and Supplementary Fig. 1). PEI was selected as the initial NP block because it facilitates the cellular internalization of NPs and their subsequent escape from endosomes[11,12], while DMNC was selected because it responds rapidly to light and its degradation products are relatively non-cytotoxic[13]. PEI–DMNC was then added to dextran sulfate (DS) to form NPs by electrostatic (PEI:DS) and hydrophobic (DMNC:DMNC) interactions. To stabilize the NP formulation, zinc sulfate was added[12,14].

NPs with an average diameter of $108.1 \pm 9.9$ nm and a zeta potential of $27.4 \pm 1.6$ mV were obtained.

To demonstrate that the NP formulation could be photo-disassembled, a suspension of NPs was exposed to UV light for up to 10 min. The number of NPs (as assessed by Kcps) decreased to below half of the initial number after 1 min of UV exposure, confirming NP disassembly (Fig. 1b). The disassembly was likely due to a change in the NP's hydrophilic/hydrophobic balance after DMNC photo-cleavage. Importantly, a conventional blue laser (405 nm, 80 mW) (Supplementary Fig. 2B) or a blue confocal laser (405 nm, 30 mW) (Supplementary Fig. 2C) can replace the UV light to induce the photo-disassembly of NPs. The response of the NPs to a blue laser was mediated by DMNC coupled to PEI, as NPs without DMNC did not respond to the laser (Supplementary Fig. 2B). We further evaluated the feasibility of triggering intracellular NP disassembly. For that purpose, HUVECs were transfected with quantum dots 525-labelled NPs for 4 h (Supplementary Fig. 2D), and a small region of the cell housing NPs was excited by a blue laser (405 nm) under a confocal microscope. Under these conditions, NP fluorescence increases compared to a reference region not excited with blue laser. This increase is due to NP disassembly and a resultant decrease in the quenching of the quantum dots immobilized in the NPs. Overall, our results show that we can disassemble NPs by light, either in vitro or within cells.

To evaluate the light-inducible polymeric NPs as a method for the controlled release of RA, a solution of RA with PEI-DMNC was assembled with DS in aqueous solution to form NPs (RA$^+$NPs). Under these conditions, the carboxyl groups of RA tend to form electrostatic interactions with the amine groups of PEI[14]. The resultant NP formulation contained approximately 150 μg of RA per mg of NP, an average diameter of 160 nm and a zeta potential of 22 mV. The release profile of the NP formulation containing [3H]RA was monitored by scintillation after NP exposure to UV light or a blue laser for up to 10 min (Fig. 1c). Up to 50 ng of RA was released per μg of NP after 10 min of activation. In the absence of light, up to 10 ng of RA was released per μg of NP after 8 days (Supplementary Fig. 3).

**Internalization and intracellular trafficking of RA$^+$NPs.** Next, we asked whether leukaemia cells could uptake light-inducible NPs. According to our dynamic laser scattering studies, RA$^+$NPs are positively charged when resuspended in different culture media and relatively stable for at least 24 h (Supplementary Fig. 4). RA$^+$NPs had no substantial effect on leukaemia cell metabolism, as evaluated by an ATP assay for concentrations up to 20 μg ml$^{-1}$ (Supplementary Fig. 5A; in case of non-leukaemic cells no cytotoxicity was observed for RA$^+$NPs up to 10 μg ml$^{-1}$—Supplementary Fig. 6). Human bone marrow APL NB4 cells and human myelomonoblastic U937 cells (expressing a chimeric PLZF/RARα fusion protein[15,16] and showing impaired sensitivity to RA[17,18]) were exposed to NPs for 4 h, washed to remove excess NPs not taken up by the cells, either exposed or not to UV light for 10 min, and finally cultured for an additional 20 h. Because the activation of RA$^+$NPs by UV or blue light can induce double-stranded DNA breaks, we evaluated the effect of this radiation on the phosphorylation of histone H2A (γH2AX), an early biomarker of DNA damage[19]. Our results indicate that neither UV nor blue light exposure for 10 min had a significant impact on DNA damage (Supplementary Fig. 5B and C).

The intracellular accumulation of RA$^+$NPs was monitored by flow cytometry and inductively coupled plasma mass spectrometry (ICP-MS) (Supplementary Fig. 7). Our results show that NP internalization peaked at 6 h (Supplementary Fig. 7A and B) and the intracellular concentration was dependent on the initial

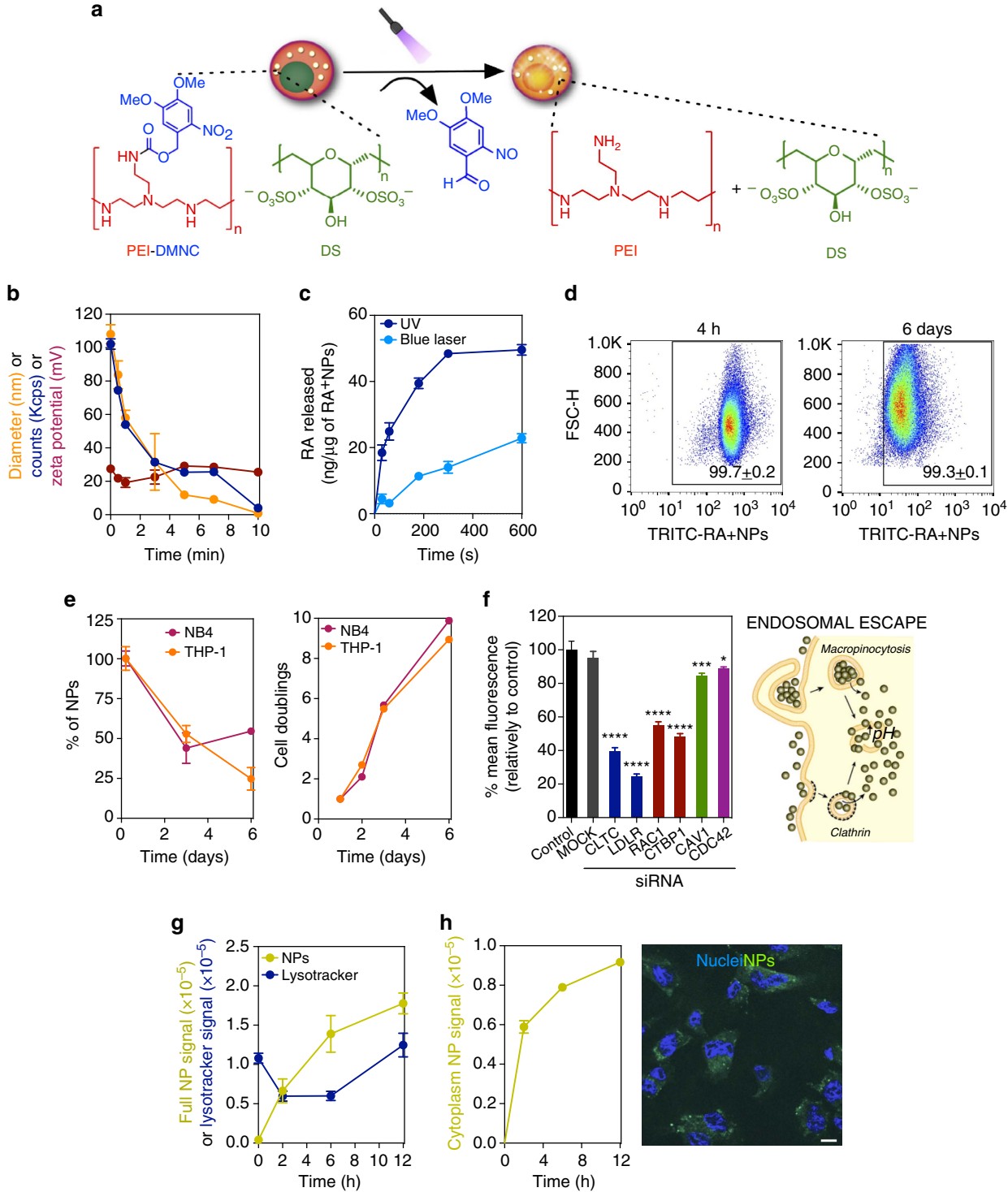

**Figure 1 | NP photo-disassembly and cellular interaction.** (**a**) Schematic representation for the photo-disassembly of RA⁺NPs. (**b**) Size, zeta potential and number of NPs (Kcps) of an aqueous suspension of light-activatable NPs (50 μg ml⁻¹) exposed to UV light (365 nm, 100 W) for up to 10 min. (**c**) Release of [3H]-RA from light-activatable NPs (10 μg ml⁻¹ in water) after exposure to UV or a blue laser (405 nm, 80 mW). (**d**) Dilution of TRITC-labelled RA⁺NPs during THP-1 cell culture as monitored by flow cytometry. Percentages of positive cells were calculated using non-transfected cells as control. (**e**) THP-1 and NB4 cells were transfected with RA⁺NPs in serum-free medium for 4 h. Left: the concentration of NPs in THP-1 and NB4 cells was monitored over 6 days by ICP-MS using Zn levels in NP. The amount of Zn in the cells after transfection (4 h) was defined as 100%. Right: cell doublings over 6 days as assessed by cell counting. (**f**) Internalization mechanism of RA⁺NPs. Left: uptake of TRITC-labelled RA⁺NPs in U937 cells after silencing key regulators of endocytosis with siRNAs. Right: schematic representation of NP internalization pathways. (**g**) Intracellular trafficking of FITC-labelled RA⁺NPs. HUVEC cells were stained with LysoTracker DND-99 to monitor NPs trafficking through the endolysosomal compartment overtime. (**h**) Fluorescence of NPs in the cytoplasm measured by confocal microscopy. Scale bar represents 10 μm. In **b,c,d,f** results are expressed as mean ± s.e.m., $n = 3$. In **g,h**, results are expressed as mean ± s.e.m., $n = 5$. In **f**, statistical analyses were performed between control and the remain experimental groups using a one-way Anova followed by a Newman–Keuls post-test. *$P < 0.05$, ***$P < 0.001$, ****$P < 0.0001$.

NP loading (Supplementary Fig. 7C). No significant differences were observed in the internalization profiles between cells. Importantly, 60–100% of the leukaemia cells retained NPs in culture for at least 6 days, as confirmed by flow cytometry (Fig. 1d and Supplementary Fig. 8). This corresponded to the retention of 25–50% of the initial mass of the NPs during 8–10 cell doublings, as obtained by ICP-MS analyses (Fig. 1e). Altogether, NPs were efficiently internalized by human leukaemia cells and accumulated intracellularly for several days.

To identify the pathways of $RA^+NP$ internalization, U937 cells were incubated in the presence of endocytosis chemical inhibitors at non-cytotoxic concentrations. Fluorescent-labelled $RA^+NPs$ were then added and the internalization monitored by flow cytometry. Whenever possible, molecules that enter by a specific internalization pathway were used as positive controls to show the efficacy of our inhibitors (Supplementary Fig. 9A). Dynasore treatment (clathrin-mediated endocytosis (CME) inhibitor) reduced the uptake of $RA^+NPs$ by 90% compared to control cells (Supplementary Fig. 9B). To further confirm the endocytosis mechanisms involved in $RA^+NP$ internalization, U937 cells were transfected with siRNAs to downregulate key components of different endocytosis pathways (Fig. 1f). We observed a ~60% and ~70% reduction in NP uptake upon downregulation of clathrin heavy chain (CLTC), and low-density lipoprotein receptor (LDLR), respectively, confirming a role for CME. The knockdown of macropinocytosis regulators Rac1 and CTBP1 led to a ~40% and ~50% decrease in $RA^+NPs$ uptake, respectively, suggesting that macropinocytosis was also involved in $RA^+NP$ internalization (Fig. 1f).

To elucidate the intracellular trafficking of our NPs, we used adherent cells (HUVECs) to allow for this characterization by confocal microscopy. The intracellular trafficking of our NPs was assessed first by performing a LysoTracker staining to see the general distribution of FITC-labelled $RA^+NPs$ in the endolysosomal system. During the first few hours, there was an increase in the intracellular signal of FITC-labelled $RA^+NPs$ and a clear drop in the intensity of LysoTracker, suggesting an increase in the pH of the endolysosomal vesicles by the presence of intracellular PEI (a strong base), as well as possible vesicle disruption, as the FITC-labelled NP signal was increased in the cytoplasm (Fig. 1g,h). At later time points (12 h) of incubation with FITC-labelled NPs, the intensity of LysoTracker reached control levels, suggesting that the endolysosomal system regained its normal characteristics. Our results further show that most of the NPs (~80% of NP fluorescence) escape endolysosomes within a few hours (Fig. 1h). At 6 h, most of the FITC-labelled NPs were distributed throughout the cytoplasm (~80% of the fluorescence) (Fig. 1h), and those that accumulated in the endolysosomal compartment were located in vesicles positive for Rab5 (early/late endosomes) and Rab7 (endosome/lysosomes) (Supplementary Fig. 10A and B). This is consistent with a rapid escape of the NPs from endosomes after uptake by the cell, likely due to their buffering capacity which gives rise to osmotic swelling and endosome rupture[12].

Next, we asked whether $RA^+NPs$ would be effluxed by leukaemia cells overtime. It is known that tumour cells express high levels of P-glycoprotein (P-gp), a membrane transporter that is responsible for the efflux of drugs[20] and NPs[21]. Therefore, we used flow cytometry to study the effects of the P-gp antagonist, verapamil[21], on the intracellular accumulation of $RA^+NPs$ in Zn-induced U937-B412 cells expressing P-gp (Supplementary Fig. 11A). Our results show that the intracellular accumulation of $RA^+NPs$ in U937 slightly decreases from 4 to 12 h (Supplementary Fig. 11B), in contrast to the control, ultra-small paramagnetic iron oxide. The intracellular accumulation of ultra-small paramagnetic iron oxide was highly dependent on the

inhibition of P-gp (Supplementary Fig. 11C). Overall, our results indicate that $RA^+NP$ endocytosis takes place primarily through a clathrin-mediated mechanism and at minor extent by macropinocytosis. It is likely that both endocytic pathways are interconnected, as has been demonstrated recently for lipid NPs[21,22]. Our results further show that the internalization of $RA^+NPs$ occurs quickly, and within the first 2 h, a significant percentage of NPs tend to escape the endolysosomal compartment, while the ones that do not escape accumulate in early/late endosomes.

***In vitro* differentiation of leukaemia cells with $RA^+NPs$.** Previously, liposomes[22] and block copolymer NPs[23] have been used for the release of RA in myeloid leukaemia cell lines. However, no significant differences were observed between liposome/NP formulations and soluble RA in their capacity to induce leukaemia cell differentiation. Therefore, we evaluated the capacity of light-inducible $RA^+NPs$ to prompt the differentiation of human $CD34^+$ primary leukaemia cells isolated from the bone marrow aspirates of acute myeloid leukaemia (AML) patients, taken at diagnosis. We evaluated their clonogenic potentials in short-term (blast colony-forming units, CFUs) and long-term (long-term culture-initiating cell, LTC-IC) assays following treatment with RA or $RA^+NPs$ (Fig. 2a). Our results indicate that light-inducible $RA^+NPs$ were more effective than soluble RA at decreasing the number of CFUs (Fig. 2b). Our results further indicate that $RA^+NPs$ activated by light are more effective in decreasing the number of CFUs (Fig. 2c) and LTC-ICs (Fig. 2d) than non-activated $RA^+NPs$. We extended the previous studies to different leukaemia cell lines, including U937, NB4 and THP-1 (Fig. 2e,g and Supplementary Fig. 12B). The differentiation process was monitored by the expression of myeloid maturation marker CD11b, a protein involved in the regulation of leukocyte adhesion and migration. For equivalent concentrations of RA, U937 (Fig. 2e) or NB4 (Fig. 2g) cells treated with $RA^+NPs$ more readily differentiated into myeloid cells than cells treated with soluble RA, likely due to the higher efficiency of the NPs to deliver RA intracellularly (Fig. 2j and Supplementary Fig. 12A). The differences in the intracellular accumulation of RA are likely explained by the differences in RA uptake mechanism. The RA is a hydrophobic small molecule with limited water solubility ($63 \, mg \, l^{-1}$ (ref. 24)) and thus *in vivo*, it is transported by proteins called cellular retinoic acid-binding proteins (CRABP) both in extracellular and intracellular compartments[25]. The expression of CRABP is relatively low in undifferentiated leukaemic cells[26] and thus the transport of soluble RA to inside of the cell might be limited. In contrast, the transport of $RA^+NPs$ occurs by endocytosis which might facilitate the intracellular accumulation of RA.

U937 (Fig. 2f) or NB4 (Fig. 2h) cells treated with light-activated $RA^+NPs$ showed a higher capacity for myelocytic differentiation than cells treated with non-activated $RA^+NPs$. In the case of NB4 cells, RA released from light-inducible $RA^+NPs$ promoted a higher percentage of cells expressing $CD13^{high}CD11b^+$ and $CD13^{low}CD11b^+$ (Fig. 3a,b) than soluble RA. NB4 cells differentiated by light-inducible $RA^+NPs$ have a different morphology from undifferentiated cells, showing polylobular nuclei, decreased nuclear:cytoplasmic ratios and decreased cytoplasm staining, as characterized by a May-Grunwald Giemsa staining (Fig. 3c). The efficiency of the $RA^+NPs$ in delivering RA inside NB4 cells and inducing a RA-dependent signalling pathway was also confirmed by the use of an NB4-RARE reporter cell line. The RA-dependent induction of a RARE element driving the transcription of the firefly luciferase gene was used to evaluate the kinetics of RA-induction using $RA^+NPs$ or RA in solution.

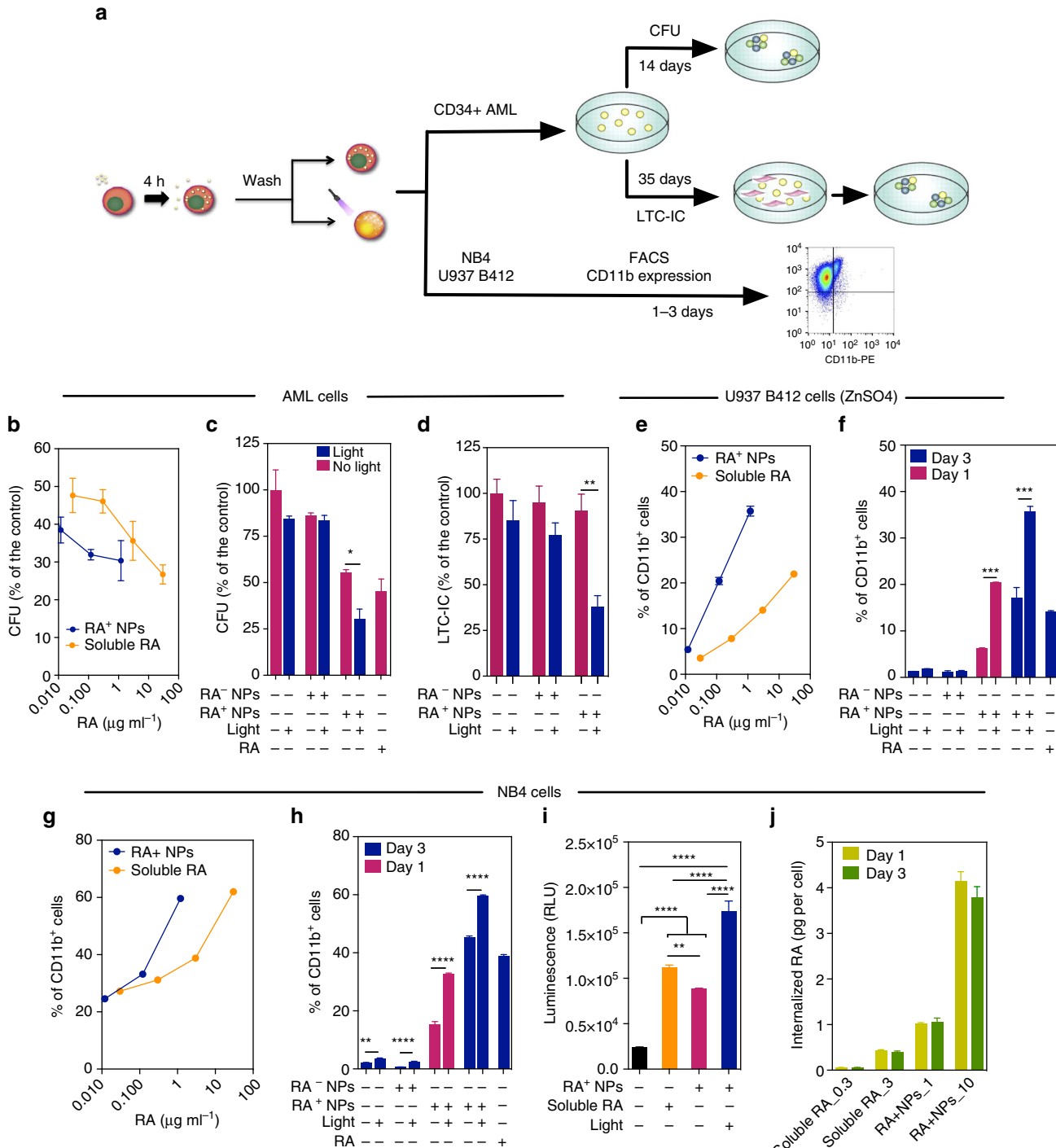

**Figure 2 | Effect of RA⁺NPs on human leukaemia cells. (a)** Schematic representation of the methodology used. **(b)** Differentiation of CD34⁺ AML cells isolated from the bone marrow aspirates of patients with AML cultured with light-activated RA⁺NPs or soluble RA. **(c,d)** Differentiation of CD34⁺ AML cells with RA⁺NPs (10 µg ml⁻¹) or blank (RA-NPs) NPs (10 µg ml⁻¹), exposed or not to UV light, by a CFU **(c)** or LTC-IC **(d)** assays. Results are expressed as a mean percentage of control plates containing only AML cells. **(e,f)** Myelocytic differentiation of human Zn-induced U937-B412 cells cultured with light-activated NPs or soluble RA at day 3 **(e)** or cultured with RA⁺NPs with or without light activation **(f)**. **(g,h)** Myelocytic differentiation of human NB4 cells cultured with light-activated NPs or soluble RA at day 3 **(g)** or cultured with RA⁺NPs with or without light activation for 1 or 3 days **(h)**. **(i)** Intracellular release of RA as evaluated by a RARE luciferase cell line. NB4-RARE cells were cultured with soluble RA (3 µg of RA per ml) for the entire duration of the experiment, or light-activatable RA⁺NPs (5 µg ml⁻¹; 0.6 µg of RA per ml). Cells were exposed to NPs for 1 h, washed with PBS and resuspended in cell media. Some samples were exposed to UV light for 5 min. The cells were then cultured for 24 h before luciferase luminescence reading. **(j)** [3H]-RA uptake by NB4 cells. NB4 cells were cultured with soluble [3H]-RA (0.3 and 3 µg ml⁻¹) for the entire duration of the experiment, or light-activatable [3H]-RA⁺NPs (1 and 10 µg ml⁻¹) for 4 h and then the cells were washed and cultured in cell medium for additional 20/68 h before scintillation counting. From **b** to **j**, results are expressed as mean ± s.e.m. (n = 3). In **c,d,f,h**, statistical analyses were performed by an unpaired *t*-test while in **i** were performed by one-way ANOVA followed by a Newman–Keuls post-test. *P < 0.05, **P < 0.01, ***P < 0.001 and ****P < 0.0001.

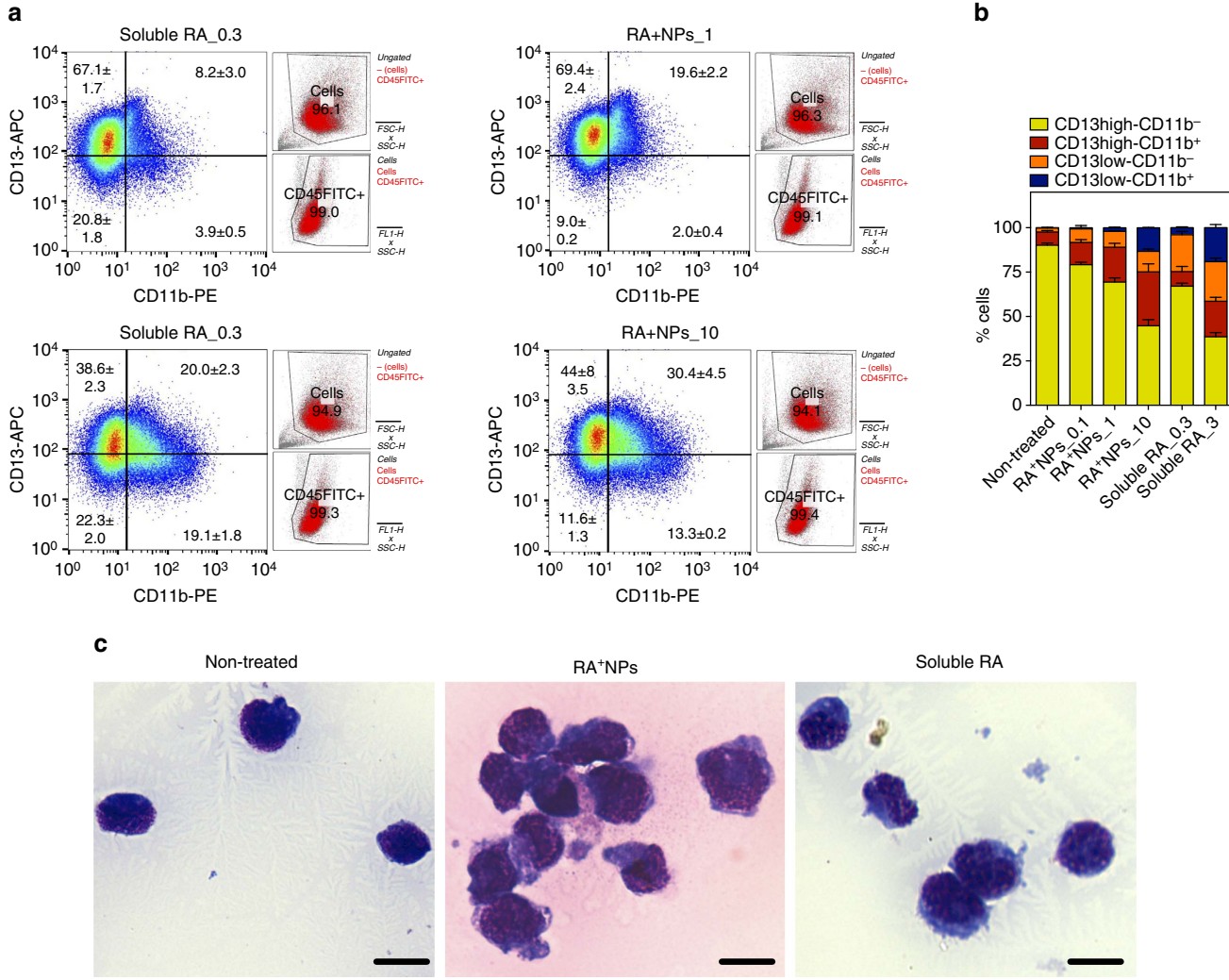

**Figure 3 | Differentiation profile of NB4$^+$ cells after treatment with soluble RA or RA$^+$NPs.** NB4 cells were incubated in serum-free RPMI-1640 with RA$^+$NPs (1 or 10 μg ml$^{-1}$; RA$^+$NPs_1 and RA$^+$NPs_10, respectively) for 4 h. Then, cells were washed three times with PBS to remove NPs not internalized, activated by UV light (365 nm, 100 W) during 5 min, and cultured in complete medium for additional 3 days. Alternatively, NB4 cells were cultured in complete medium for 3 days having soluble RA (0.3 or 3 μg ml$^{-1}$; RA_0.3 and RA_3, respectively). (**a**) Flow cytometry analysis of differentiated NB4 cells cultured for 3 days. Cells were initially gated in an FSC/SSC plot (in red; attached to the scatter plot) and then gated for the expression of CD45 (in red; attached to the scatter plot). Percentages of positive cells were calculated based in the isotype controls and are shown in each scatter plot. Results are expressed as mean ± s.e.m. (n = 3). (**b**) Percentages of CD13$^{high}$CD11b$^-$, CD13$^{high}$CD11b$^+$, CD13$^{low}$CD11b$^-$ and CD13$^{low}$CD11b$^+$ in NB4 cells differentiated for 72 h. Results are expressed as mean ± s.e.m. (n = 3). (**c**) Representative May-Grunwald Giemsa stains of NB4 cells cultured for 72 h after exposure to RA$^+$NPs (10 μg ml$^{-1}$; therefore 1.2 μg ml$^{-1}$ of RA) or soluble RA (3 μg ml$^{-1}$). Bar corresponds to 20 μm.

Our results showed that RA$^+$NPs were able to induce high levels of luciferase activity shortly after light activation (Fig. 2i). RA$^+$NPs were more efficient and quicker at inducing transcription from the RARE-luciferase locus than RA in solution.

Next, we monitored the effects of soluble RA and RA released from RA$^+$NPs in the transcriptional regulation of 96 genes involved in the differentiation program of NB4 and U937 cells[27,28] (Supplementary Figs 13 and 14). In the case of NB4 cells, CD52, CDKN1A (also known as the p21 gene) and TNFα are recognized as direct targets of RA[27]; CD52 and CDKN1A are upregulated and TNFα is downregulated in response to RA. Our results show that the upregulation of CDKN1A and CD52 mRNA transcripts was significantly higher in cells treated with RA$^+$NPs compared to those treated with soluble RA for 8 h and 24–48 h, respectively (Supplementary Fig. 15). On the other hand, the downregulation of TNFα occurred more slowly in cells treated with RA$^+$NPs than in those treated with soluble RA. Other RA-responsive genes were also evaluated by qRT–PCR[27,29].

The number of mRNA transcripts for IL-8 and Ly6E genes was higher in cells treated with RA$^+$NPs than in those treated with soluble RA for 8–24 h and 24–48 h, respectively. In addition, the number of mRNA transcripts for CCR2 and CEBPα genes was lower in cells treated with RA$^+$NPs than in those treated with soluble RA for 8–48 h and 8 h, respectively. Importantly, no significant variation in mRNA transcripts for RARα, RARβ and RARγ (Supplementary Figs 13 and 15) was found between all experimental groups. Therefore, our results suggest that the differentiation profile induced by RA$^+$NPs is similar to that observed for soluble RA; however, the magnitude of the effect in the initial stages (up to 24 h) is higher in cells treated with RA$^+$NPs.

Recent studies report that lipid-based NPs are able to release siRNA cargo in the cytosol only during a limited window of time, when the NPs reside in a specific compartment that shares early and late endosomal characteristics[30,31]. Therefore, we evaluated the impact of temporal activation of RA$^+$NPs on the

differentiation of NB4 and U937 cells. The effect of the intracellular release of RA was evaluated in terms of cell differentiation into the myeloid lineage (expression of CD11b at day 3) (Supplementary Fig. 16A). Cells that were incubated with RA$^+$NPs for 4–8 h and subsequently activated by light showed the highest differentiation into the myeloid lineage. Interestingly, cells incubated with RA$^+$NPs for 1 h showed an already high level of differentiation, indicating that NP uptake is rapid and their endosomal escape is efficient. Then, we asked whether cells exposed for the same time (4 h) to NPs but activated at different times (up to 44 h) during the intracellular trafficking of RA$^+$NPs would have differences in their differentiation pattern (Supplementary Fig. 16B). Our results show that even after 44 h following NP internalization, the NPs can be activated and the resultant delivery of RA in the cytosol can be relatively efficient. Similar results were obtained when the effect of temporal intracellular RA release was evaluated by the induction of the NB4-RARE reporter cell line (Supplementary Fig. 16B). Overall, these results show that the NPs can be accumulated in the intracellular environment and activated after a few days, without losing their capacity to promote cell differentiation. Importantly, a single activation procedure seems to be sufficient to activate RA release from the NPs because no significant increase in efficiency was observed when multiple rounds of light activation were performed (Supplementary Fig. 17).

***In vivo* differentiation of leukaemia cells with RA$^+$NPs**. Next, we evaluated if RA$^+$NPs can function *in vivo*. We investigated whether leukaemic cells loaded with RA$^+$NPs could be activated *in vivo* after subcutaneous transplantation. We selected a Matrigel plug subcutaneous model because we know what is the attenuation of the blue laser through the murine skin (Supplementary Fig. 18A) and we could easily remove the cells from the Matrigel plug and evaluate their differentiation by flow cytometry. Initially we evaluated whether the *in vivo* environment could interfere with the differentiation program of leukaemic cells after *ex vivo* activation. Human NB4 cells were cultured with RA$^+$NPs for 4 h, washed and activated *ex vivo* by exposure to a 405 nm blue laser (80 mW) for 5 min, embedded in Matrigel and then injected into a cylindrical poly(dimethylsiloxane) (PDMS) construct that was previously implanted subcutaneously in NOD/SCID recipients (Fig. 4a). The PDMS cylinder was used to restrict cell position inside the animal. After 5 days, human cells were isolated from the implants and CD11b expression measured by flow cytometry (Fig. 4b). Consistent with our *in vitro* data, CD11b expression was statistically higher in NB4 cells treated *ex vivo* with light-activated RA$^+$NPs than in cells treated with RA$^+$NPs without light activation (Fig. 4b). The experiment was repeated with *in vivo* activation. One day after implantation, the recipients were exposed to a 405 nm blue optical fibre for 5 min at the implant site (Fig. 4c). After 3 days, the recipients were killed, the human cells were isolated and CD11b expression was assessed. CD11b expression was higher in NB4 cells from mice that had been exposed to the blue laser, demonstrating that internalized RA$^+$NPs can be activated to release RA *in vivo* in a highly controlled manner (Fig. 4c).

**Homing and differentiation of leukaemia cells with RA$^+$NPs**. Leukaemia cells that resist therapy reside in microenvironmental niches in the bone marrow that are difficult to reach by conventional therapy[6]. Therefore, we investigated whether leukaemia cells transfected with RA$^+$NPs could engraft into the bone marrow and localize at nearby resident leukaemia cells. For this purpose, we transplanted leukaemia cells into immunodeficient CB17-Prkdcscid/J (NOD/SCID) mice treated with anti-CD122 to

deplete innate immune cells (NS122). Green fluorescent protein (GFP)-labelled THP-1 cells, a human APL cell line (NB4 cells were unable to engraft), were injected into sub-lethally irradiated mice in the tail vein, followed by the injection of THP-1 cells transfected with RA$^+$NPs after 24 h (Fig. 5a). After 1–2 days, animals were killed and the cells in the animal's calvaria (cranium) were characterized by *ex vivo* imaging (Fig. 5b). Both cells (GFP-labelled and NP-labelled THP-1 cells) could engraft in the bone marrow of the animals and were detected after 1 or 2 days. Therefore, NP-labelled THP-1 cells (delivery cells) could enter into the same niche as the GFP-labelled cells (target cells) (Fig. 5c).

Next, we investigated whether we could differentiate the NP-labelled human THP-1 cells (delivery cells) at the bone marrow niche (more specifically at the calvaria), and whether they could modulate nearby cells (Fig. 6a). Blue laser attenuation studies in the calvaria bone using a photometer showed an attenuation of ≈75% (Supplementary Fig. 18B). Human THP-1 cells transfected with RA$^+$NPs were injected intravenously into CB17-Prkdcscid/J (NOD/SCID) mice, and 6 days after injection, the cranium was exposed to a blue laser for 5 min. Seventy-two hours after blue laser irradiation, the engraftment of human CD45$^+$ cells in the long bones of non-irradiated and blue laser-irradiated mice was 47.7 ± 13.0 and 39.8 ± 3.5%, respectively (Fig. 6b). No statistical difference was found for the engraftment in both groups. *Ex vivo* staining of mice calvaria showed that human CD45$^+$ cells in the non-irradiated mice did not express CD11b, while ∼80% of the human CD45$^+$ cells in blue-irradiated mice expressed CD11b (Fig. 6c–e). In addition, cells surrounding CD45$^+$CD11b$^+$ cells in irradiated mice expressed CRABP, a RA protein transporter, while negligible levels of CRABP were found in non-irradiated mice (Fig. 6f). Therefore, THP-1 cells differentiated by a blue laser had the capacity to modulate the cells in the vicinity. This is likely due to RA secreted (including exosomes containing RA) by the THP-1 cells, as observed *in vitro* using the NB4 cell line (Supplementary Fig. 19). Overall, our results show that leukaemia cells transfected with RA$^+$NPs could (i) engraft into the bone marrow and localize near to resident leukaemia cells, (ii) differentiate after blue laser activation and (iii) modulate the activity/phenotype of the resident leukaemia cells. It is possible that the cells that reach the bone marrow are the ones that are less affected by the passive diffusion of RA (that is, RA release without light activation) within the cells. Therefore, the cells that engraft are likely the ones that have less leaky RA$^+$NPs and thus are undifferentiated leukaemic cells (CD11b$^{negative}$ phenotype).

**Discussion**. In conclusion, we have developed a light-inducible polymeric NP for the spatio-temporal release of RA within leukaemia cells. We showed the efficiency of RA release both *in vitro* and *in vivo*. The efficiency is due to a combination of several factors, including (i) high concentration of intracellular RA, (ii) high endolysosomal escape (80% of the NPs escape the endolysosomal compartment in the first 5 h), (iii) prolonged intracellular accumulation of the NPs (more than 6 days in leukaemia cells) and (iv) rapid disassembly of the NPs once activated by UV or blue light (within minutes). We further showed that cells transfected with light-inducible NPs can be activated after six days while maintaining the same inductive properties. This gives an opportunity to use cells for activation at specific sites in the human body and differentiate cells in their proximity. Although the present study uses a RA delivery system as a proof of concept, a similar approach could be implemented for other drugs relevant to therapeutic or regenerative medicine.

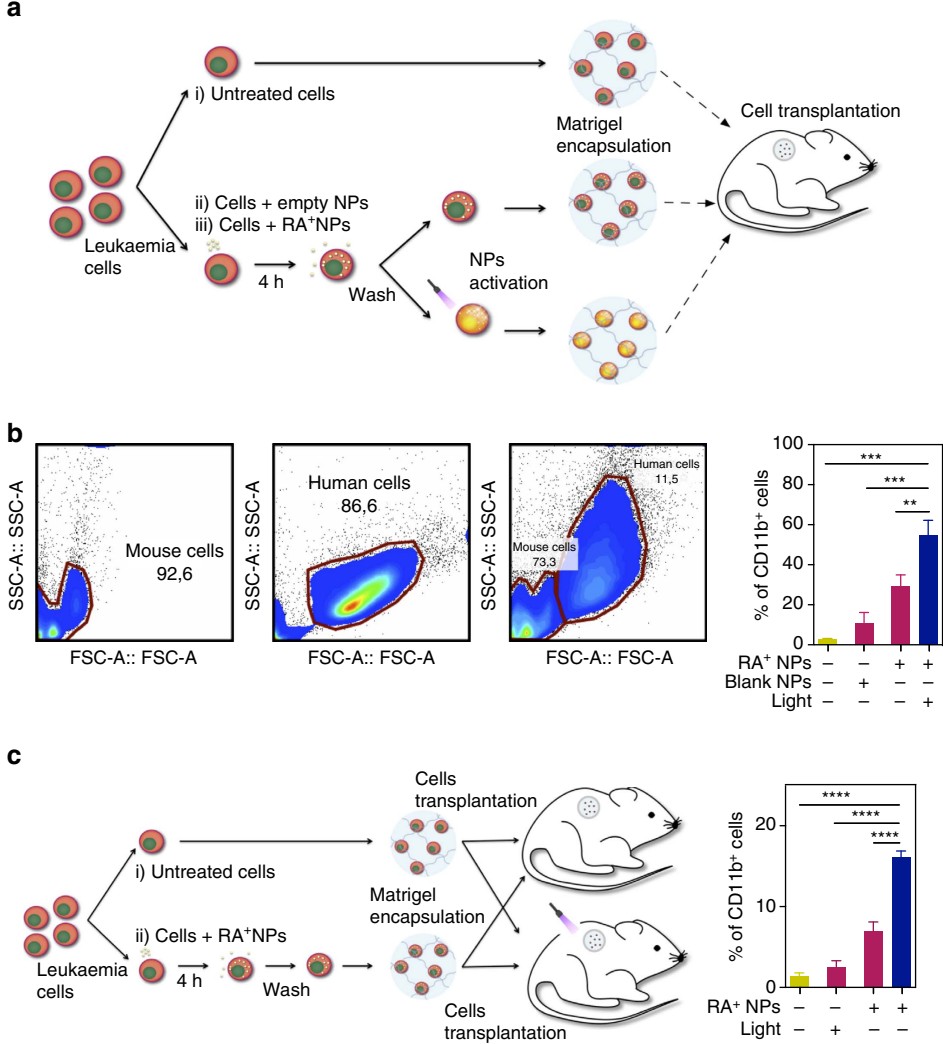

**Figure 4 | *In vivo* differentiation of NB4 cells exposed to light-activatable RA⁺NPs.** (**a**) Schematic representation of the *in vivo* experimental set up. Cells were treated with blank or RA⁺NPs (both at 10 µg ml⁻¹) for 4 h, washed and then activated or not with a blue optical fibre (405 nm, 80 mW) for 5 min. Cells were then resuspended in a 1:1 (v/v) Matrigel solution and subcutaneously injected in a PDMS cylinder construct implanted in the dorsal region of mice. After 5 days, cells were removed from the construct and characterized by flow cytometry, for CD11b expression. (**b**) Representative flow cytometry plots for mice recipient cells (left), human leukaemia NB4 cells (middle) and a mixture of mice recipient cells with human leukaemia NB4 cells (right) and percentage of CD11b⁺ cells in human leukaemia NB4 cells collected 5 days after subcutaneous injection. (**c**) Schematic representation of the *in vivo* experimental set up. Cells were treated with RA⁺NPs (10 µg ml⁻¹) for 4 h, washed and then encapsulated in a 1:1 (v/v) Matrigel solution and subcutaneously injected in a PDMS cylinder construct implanted in the dorsal region of mice. After 24 h, some experimental groups were activated *in vivo* with a blue optical fibre for 5 min. Plot shows the percentage of CD11b⁺ cells in human leukaemia NB4 cells collected 3 days after the *in vivo* activation. In **b,c**, results are expressed as mean ± s.e.m. (*n* = 3-4). Statistical analyses were performed by one-way Anova followed by a Newman–Keuls post-test. **P < 0.01, ***P < 0.001, ****P < 0.0001.

The present research reports a blue laser/UV-inducible NP. The platform was designed for initial proof of concept regarding the modulation of leukaemic cell niches in the bone marrow. Recent studies have explored UV/blue laser approaches to trigger *in vivo* the presentation of bioligands with spatial-temporal control to regulate cell adhesion, inflammation and vascularization of biomaterials[32] and to detect protease activity at sites of disease[33]. However, the *in vivo* applications of this light trigger are limited, as the optical properties of biological tissue cause attenuation and scattering of light rays, reducing their penetration depth. The recent advances in developing NPs activated by near-infrared light[34] may provide a better option for enhancing light penetration in tissues. For example, upconversion nanocrystals that convert near infrared light into visible light (blue light) may

be an interesting possibility for the photo-cleavage of linkers to which RA is chemically bound[35].

We have used leukaemic cells as a programmable delivery system that recognizes the leukaemic niches and modulate them after light activation. The aim of the study was to show the feasibility to remotely induce the differentiation of leukaemic cells in the bone marrow and not to demonstrate the exhaustion of cancer cells, as this will be difficult using the blue laser activation process. The concept here is different from previous studies that have used therapeutic NPs coated with aptamers to direct them to the bone marrow[36]. Although the results achieved are interesting, the side effects of the systemic administration of the NPs containing a powerful anti-leukaemic agent remain to be determined. In the present study we have used cancer biology

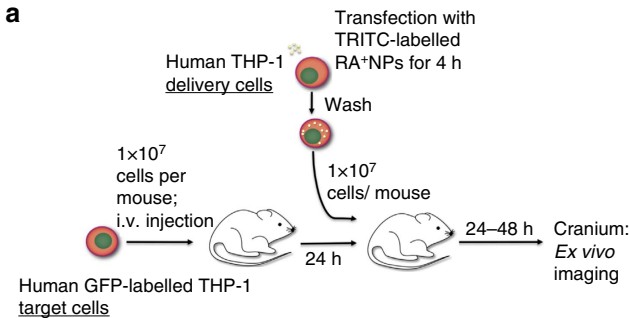

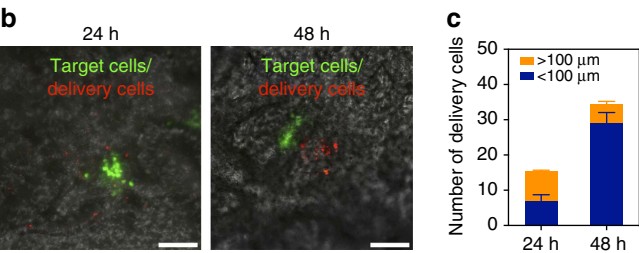

**Figure 5 | Leukaemia cells transfected with RA⁺NPs can home at leukaemia niches.** (**a**) Schematic representation of the protocol. Human GFP-labelled THP-1 cells were employed as the target cells and human THP-1 cells without GFP label were employed as delivery cells. On day 0, the target cells were injected into the NOD/SCID mice by intravenous injection through the tail vein. On day 1, the delivery cells loaded with RA⁺NPs were injected intravenously through the tail vein. After 24 ($n = 3$) or 48 h ($n = 3$) of the delivery cell injection, the mice were killed, the craniums collected and examined with a confocal microscope. (**b**) Both target and delivery cells were observed in mice craniums at 24 and 48 h. Bar scale means 100 µm. (**c**) Delivery cells close (<100 µm) or distant (>100 µm) to target cells were quantified. According to our results, the delivery cells can enter into the same niche as the target cells. Results are expressed as mean ± s.e.m. ($n = 3$).

to fight cancer. We have used leukaemia cells loaded with a clinical-approved AML differentiating agent able to home to leukaemic niches in the bone marrow. Although not explored in the current study, normal hematopoietic stem cells may also be used as a nanoparticle carrier as they are able to home to leukaemic niches[33].

## Methods

Detailed methods are available in Supplementary Information.

**Preparation and characterization of PEI conjugated with DMNC.** DMNC (48.5 mg, Sigma) was slowly added to a solution of PEI in DMSO (2 ml containing 50 mg ml⁻¹ PEI, Sigma) containing triethylamine (24.5 µl, Sigma), and the reaction flask cooled to 0 °C by immersion on ice. Then, the reaction was allowed to proceed for 24 h at 25 °C with stirring. At the end, the PEI-DMNC conjugate was purified by dialysis (Spectra/Por 1 Regenerated Cellulose dialysis membrane, MWCO 6000-8000 Da, Spectrum) against DMSO overnight at room temperature. Reaction yields above 54% were obtained. For NMR characterization, PEI-DMNC (in DMSO) was precipitated in water, washed, freeze-dried and then dissolved (10 mg ml⁻¹) in DMSO-d6 and ¹H NMR spectra were acquired using a Bruker Avance III 400 MHz spectrometer.

**Preparation of RA⁺NPs.** For the preparation of RA⁺NPs, a RA solution (24 µl, 50 mg ml⁻¹, in DMSO) was added to a solution of PEI-DMNC (66.7 µl, 150 mg ml⁻¹ in DMSO) and maintained at room temperature for 30 min, under stirring. The solution was then carefully added to an aqueous solution of DS (5 ml, 0.4 mg ml⁻¹) and stirred for 5 min. The NPs in suspension were treated with an aqueous solution of ZnSO₄ (120 µl; 1 M) for 30 min. RA that was not encapsulated in the NPs was removed by centrifugation (12,000g for 3 min). The NP suspension was then dialysed (Spectra/Por 1 Regenerated Cellulose dialysis membrane, MWCO 6,000–8,000 Da, Spectrum) for 24 h, in the dark, against an

aqueous solution of mannitol (5%, w/v), lyophilized for 1 day and stored at 4 °C before use.

**Cell culture.** HUVECs (Lonza) were cultured in EGM-2 medium (Lonza) in a CO₂ incubator at 37 °C, 5% CO₂ in a humidified atmosphere, with media changes performed every other day. Cells were passaged every 2–5 days and used for experiments between passages 4 and 6. Human bone marrow APL NB4 cells, kindly provided by Dr Arthur Zelent (Institute of Cancer Research, Royal Cancer Hospital), were cultured in RPMI-1640 (Gibco) supplemented with 10% fetal bovine serum (FBS) (Gibco) and 100 U ml⁻¹ PenStrep (Lonza). Human myelo-monoblastic cell lines U937-MT and U937-B412, kindly provided by Dr Estelle Duprez (Centre de Recherche en Cancérologie de Marseille, France), were maintained at exponential growth in RPMI-1640 medium supplemented with 10% FBS and 100 U ml⁻¹ of PenStrep. For PLZF/RARA induction cells were stimulated with 0.1 mM ZnSO₄ for at least 24 h. Human acute monocytic leukaemia cell line THP-1 (DSMZ no. ACC16) was cultured in RPM1-1640 media supplemented with HEPES (10 mM), sodium pyruvate (1.0 mM) and 2-mercaptoethanol (0.05 mM).

**NP internalization studies.** NP internalization was monitored by ICP-MS. In this case, the intracellular levels of Zn were measured before and after cell exposure to NPs. NB4 and U937 cells (0.1 × 10⁶ cells per well) were plated in 24-well plates and incubated in serum-free RPMI-1640 from 1 to 24 h with variable amounts of RA⁺NPs. After incubations, NPs that were not internalized by the cells were washed (three times with PBS) and the cells were centrifuged; followed by the addition of an aqueous solution of nitric acid (1 ml, 69% (v/v)). The samples were analysed by ICP-MS for the concentration of intracellular levels of Zn. The concentration of Zn was normalized per cell. The estimation of NPs was done based on controlled standard solutions.

**Mechanism of NP uptake.** U937 cells were cultured on 24-well plates (1 × 10⁵ cells per well) and inhibited by one of the following chemicals during 30 min before adding a suspension of TRITC-labelled NPs (5 µg ml⁻¹): EIPA (50 µM), dynasor (80 µM), dansylcadaverine (100 µM), cytochalasin D (10 µM), nocodazole (50 µM), filipin III (100 µM) and polyinosinic acid (100 µg ml⁻¹). The inhibitor concentrations were based in values reported in literature and further validated by us to have no cytotoxic effect over the period of the assay (6 h), as confirmed by an ATP assay. The incubation of the cells with NPs for different times was performed in the presence of the inhibitor. As controls, we used cells without NPs and cells incubated with NPs without inhibitor. At the end of each time point, cells were centrifuged at 1,300 r.p.m., 20 °C for 5 min with PBS, washed one time with cold trypan blue solution (200 µl; 600 µg ml⁻¹), re-washed three times with cold PBS and then resuspended in PBS containing 2.5% FBS (500 µl) for flow cytometry analysis. A total of 10,000 events were obtained per measurement. To validate the inhibitory activity of dynasor we performed uptake studies of FITC-labelled transferrin, known to selectively enter cells via clatherin-mediated endocytosis. Briefly, U937 cells were cultured on 24-well plates (1 × 10⁵ cells per well) and treated or not with dynasor (80 µM, 30 min pre-incubation), followed by addition of 1 µg ml⁻¹ FITC-labelled transferrin (Life Technologies). The transferrin was allowed to bind for 3 min at 4 °C. Cells were then evaluated as before.

The NP uptake mechanism was also studied on U937 cells by silencing specific proteins of CME (CLTC and LDLR), caveolin-mediated endocytosis (CAV1), GEEC-CCLIC pathways (CDC42) and macropinocytosis (RAC1 and CTBP1) by siRNA (Thermo Fisher). Transfection was performed in a 24-well plate with 0.5 × 10⁵ cells in antibiotic-free complete medium with 100 nM siRNA and 1.5 µl of Lipofectamine RNAiMAX (Life Technologies) transfection reagent for 24 h. After this initial period, the transfection medium was replaced by complete medium and the cells incubated for another 48 h. Then, cells were cultured with TRITC-labelled NPs (5 mg ml⁻¹) for 6 h. Once the incubations were terminated, the cells were centrifuged at 1,300 r.p.m., 20 °C for 5 min, with PBS, washed one time with cold trypan blue solution (200 µl; 600 µg ml⁻¹), re-washed three times with cold PBS and then resuspended in PBS containing 2.5% FBS (500 µl) for flow cytometry analysis. Non-transfected cells or cells transfected with lipofectamine but without siRNAs (MOCK) were used as controls. In all flow cytometry analysis, a total of 10,000 events were recorded per run. All conditions were performed in triplicate.

**Intracellular trafficking analyses of NPs.** HUVEC cells (passage 4) were cultured on 1% gelatin-coated slides until subconfluency in EGM-2. The cells were then incubated with FITC-labelled NPs (1 µg ml⁻¹) for 1 or 4 h, washed extensively, exposed or not to UV light (365 nm, 100 W) and cultured in normal conditions for up to 12 h. For LysoTracker staining, the cells were incubated with LysoTracker Red DND-99 (50 nM, Invitrogen). After 30 min of incubation, the coverslips were washed extensively with PBS, followed by cell fixation with paraformaldehyde (4%, Electron Microscopy Sciences) for 10 min at room temperature and then washed with PBS. Cell nuclei were stained with 4′,6-diamidino-2-phenylindole (DAPI) (Sigma), and the slides were mounted with mounting medium (Dako) and examined with a Zeiss LSM 50 confocal microscope.

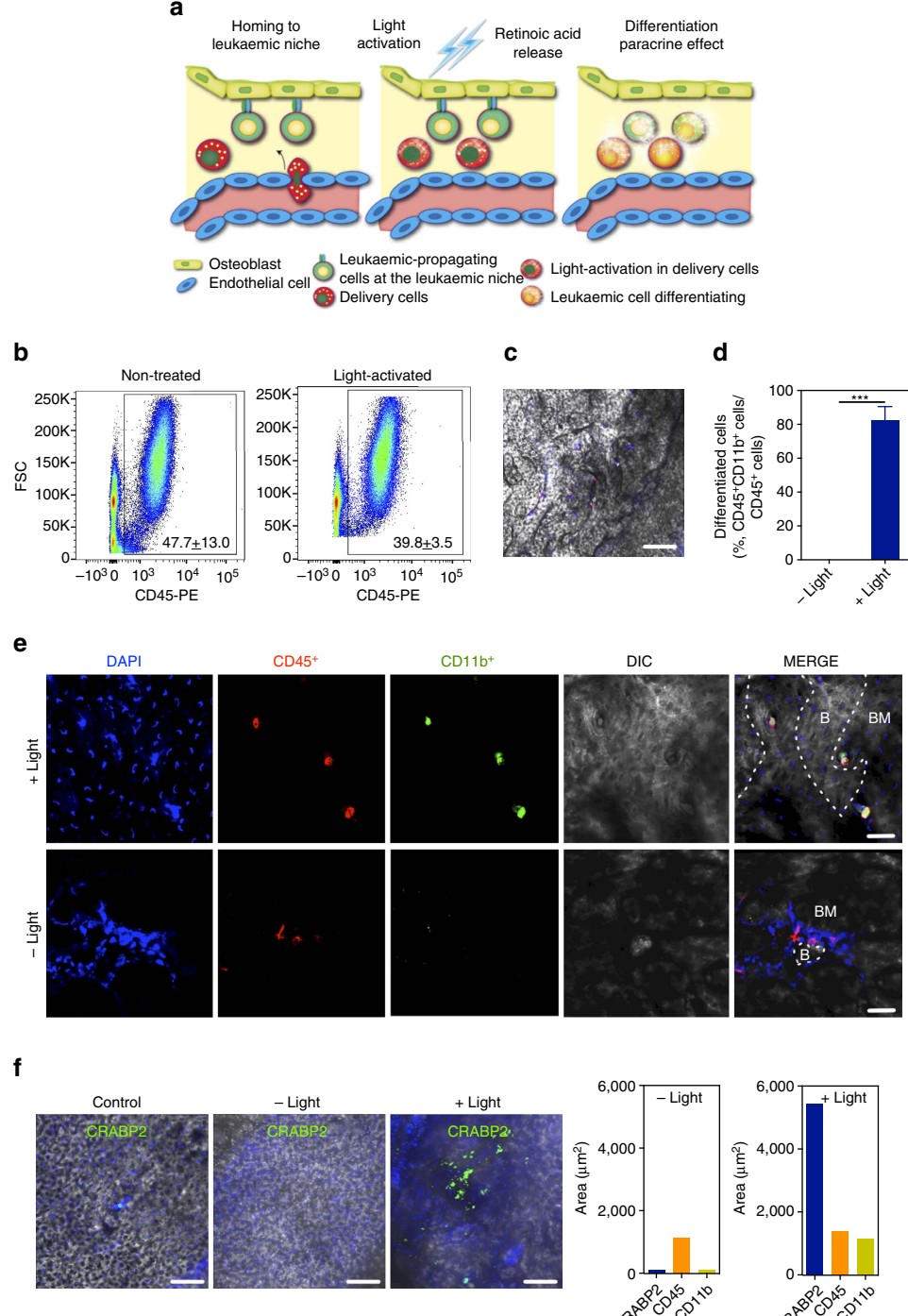

**Figure 6 | *In vivo* activation of leukaemic THP-1 differentiation program by light at the bone marrow.** (**a**) Schematic representation for the activation of human THP-1 cells transfected with RA + NPs in the bone marrow niche leading to the modulation of leukaemia cells. (**b**) Percentage of human CD45 + cells in the long bones as evaluated by flow cytometry. NOD/SCID mice were injected intravenously with THP-1 cells transfected with RA + NPs through the tail vein. After 6 days, animals (*n* = 6) were either irradiated by a blue laser (5 min, 405 nm, 80 mW) or not in the calvaria (*n* = 6). Three days after irradiation, mice were killed and the long bones were collected. Cells in the long bones were characterized by flow cytometry for the expression of human CD45 epitope. The engraftment of THP-1 cells was slightly higher in the non-treated than light-activated animals, but not statistically different. Percentages of positive cells were calculated based in the isotype controls and are shown in each scatter plot. Results are mean ± s.e.m. (*n* = 3). (**c–e**) Differentiation of human CD45 + cells into CD11b + cells, as evaluated by immunofluorescence, in the calvaria of animals irradiated or not with a blue laser. B, bone; BM, bone marrow. In **d**, results are mean ± s.e.m. (*n* = 3). ***P < 0.001. (**f**) Expression of CRABP2 in cells of calvaria of animals irradiated or not with a blue laser. Scale in **c,e,f** means 200, 50 and 100 μm.

**NP dilution during cell proliferation.** NP dilution with cell growth was monitored over 6 days by ICP-MS by the quantification of intracellular levels of Zn. NB4 and THP-1 cells ($0.5 \times 10^6$ cells per ml) were plated in six-well plates and incubated in serum-free RPMI-1640 with 20 μg ml$^{-1}$ of RA + NPs. After 4 h incubation, NPs that were not internalized by the cells were washed three times with PBS and the cells were left to grow at $0.2 \times 10^6$ cells per ml in complete medium for additional

4 h, 3 days and 6 days, maintaining always an exponential growth. After each incubation, cells were counted, collected by centrifugation and resuspended in nitric acid (1 ml, 69% (v/v)) for ICP analysis. The concentration of Zn was normalized per cell. The estimation of NPs was done based on Zn quantification in 20 µg of NPs. In some experiments, cells were transfected with RA$^+$NPs labelled with TRITC, and their fluorescence monitored by flow cytometry overtime, to evaluate NPs distribution within the cells.

**[³H]RA internalization studies.** [11, 12-³H(N)]-Retinoic acid, 50.4 Ci mmol$^{-1}$, was purchased from Perkin Elmer. [³H]RA solution for cell culture assays was prepared on the day of experiments by dissolving [³H]RA in DMSO with unlabelled RA in a 1:1,000 ratio to a final concentration of 10 µM of RA. [³H]RA solution in DMSO for the preparation of NPs was prepared on the day of experiments using a 1:4,000 ratio of labelled to unlabelled RA. Experiments were initiated by the adding the [³H]RA solution (1 µM and 10 µM; representing less than 1% in volume of the total cell culture medium) or [³H]RA-NP suspension (1 µg ml$^{-1}$ and 10 µg ml$^{-1}$) to cultures (60,000 cells per condition, 24-well plate, 1 ml) of NB4 or U937 cells. In case of soluble RA, cells (NB4 or U937; 60,000 cells per condition, 24-well plate) were cultured with medium containing [³H]RA (1 µM and 10 µM; 1 ml of medium) for 24 or 72 h, washed with PBS (two times), collected, lysed with lysis buffer (100 ml) and kept on ice until scintillation counting procedure. In case of RA-containing NPs, cells (same conditions as for soluble RA) were cultured with [³H]RA-NPs (1 µg ml$^{-1}$ and 10 µg ml$^{-1}$) for 4 h, washed with PBS and cultured for additional 20 or 68 h in the respective culture medium. Cells were then collected to eppendorfs, washed with PBS, centrifuged (1,500 r.p.m., 5 min), lysed with lysis buffer (see above) and kept on ice until scintillation counting procedure. The lysed samples (100 ml) were mixed with liquid scintillation fluid (1 ml; Packard Ultima Gold) and the scintillations counted in a TriCarb 2900 TR Scintillation analyser (Perkin Elmer).

**NB4 differentiation assay.** Myelocytic differentiation of NB4 cells was assessed by quantifying CD11b$^+$, CD11b$^+$CD45$^+$CD13$^{high}$ or CD11b$^+$CD45$^+$CD13$^{low}$ populations using flow cytometry. NB4 cells (between $6.0 \times 10^4$ and $10 \times 10^4$ cells per condition) were plated in 24-well plates and cultured with soluble RA (3 µg ml$^{-1}$) or light-activatable RA$^+$NPs (10 µg ml$^{-1}$) for 3 days. The NPs were suspended in serum-free medium and added to cells for 4 h. The cells were then washed by centrifugation (1,300 r.p.m., 5 min) to remove non-internalized NPs, and half of the samples were exposed to UV light (365 nm, 100 W, 5 min). The cells were then cultured up to 3 days in RPMI-1640 medium supplemented with 10% FBS and 100 U ml$^{-1}$ PenStrep with half medium changes every 3 days.

**U937 differentiation assay.** Myelocytic differentiation of U937 cells was assessed by the quantification of CD11b expression by flow cytometry. U937-B412 cells (6.0 × 10$^4$ cells per condition) were cultured either with or without ZnSO$_4$ (0.1 mM). To induce the expression of promyelocytic leukaemia zinc finger/RARα (PLZF/RARα) in U937-B412 cells they were treated for 24 h with ZnSO$_4$ (0.1 mM). Then cells were treated with soluble RA or light-activatable RA$^+$NPs (transfection for 4 h followed by light activation for 5 min) for 3 days. After 1 and 3 days, expression of CD11b on U937 cell surface was measured by staining with a fluorescent (PE)-conjugated anti-CD11b mAb (BD Biosciences) using flow cytometry.

**AML differentiation assay.** AML bone marrow mononuclear cells isolated by Ficoll-Histopaque (GE Healthcare) gradient centrifugation, enriched using the MACS CD34 isolation kit (Miltenyi Biotec) and cryopreserved were kindly provided by Dr Rajeev Gupta (Department of Haematology, UCL Cancer Institute). The AML cells were isolated from an 85-year-old male patient with AML 34 + 117 + 33 + 13 + DR + 35% blasts and a 70-year-old woman with RAEB2/evolving AML 34 + 117 + 33 + 12% blasts. Both samples had formal karyotyping/extended FISH panels. Briefly, neither patients were PML-RARA: one was standard risk (normal karyotype with a small trisomy 8 subclone), and the other was high risk with complex karyotype. The isolated CD34$^+$ AML cells were maintained in StemSpan SFEM medium (Stemcell Technologies) supplemented with a human cytokine cocktail containing SCF (50 ng ml$^{-1}$, Stemcell Technologies), TPO (15 ng ml$^{-1}$) and Flt-3L (50 ng ml$^{-1}$, PeproTech) plus PenStrep (10,000 U ml$^{-1}$, Lonza) and Fungizone (25 mg ml$^{-1}$, Sigma) up to 3 days. Prior to the colony-forming cell (CFC) and LTC-IC assays, AML cells were incubated for 4 h in ex vivo (Lonza) serum-free medium, with and without blank NPs or RA$^+$NPs in a 24-well plate. After that time, the cells were washed to remove loosely bound NPs. For CFC assays (2.0 × 10$^5$ cells per condition), AML cells were plated in triplicate in MethoCult H4230 medium (3 ml, StemCell Technologies) supplemented with SCF (50 ng ml$^{-1}$), IL-3 (10 ng ml$^{-1}$) and Flt-3L (50 ng ml$^{-1}$), all human, plus PenStrep (10,000 U ml$^{-1}$, Lonza) and Fungizone (25 µg ml$^{-1}$, Sigma) in six-well plate. In some conditions, RA$^+$NPs accumulated within the cells were activated by a UV light (365 nm, 100 W, 5 min). Cultures were scored after 14 days for the presence of clusters and colonies containing >20 cells using an inverted microscope. LTC-IC assays were performed in triplicate in a six-well plate gelatinized for 2 h before adding the feeders. The feeder layer was composed of a 1:1 mixture of irradiated (80 Gy) SL/SL (1.5 × 10$^4$ cells per condition) and M210B4 mouse fibroblasts

(1.5 × 10$^4$ cells per condition). AML cells (1 × 10$^6$ cells per condition) were plated in Myelocult H5100 medium (StemCell Technologies), supplemented with Flt-3 L (50 ng ml$^{-1}$), hydrocortisone (10$^{-6}$ M) (StemCell Technologies) and PenStrep (10,000 U ml$^{-1}$, Lonza) and fungizone (25 mg ml$^{-1}$, Sigma). For some conditions UV light (365 nm, 100 W, 5 min) was used to trigger RA release. After the cells were inoculated, weekly half medium changes were performed (with Flt-3L (100 ng ml$^{-1}$)) for the duration of the culture. After 5 weeks, all cells were collected and placed into methylcellulose-based assay for the detection of AML-CFC as described above.

**Subcutaneous implantation in vivo.** The animal work has been conducted according to relevant national and international guidelines and approved by the Bioethics Committee of University of Salamanca. On the day before injecting the cells, PDMS cylindrical constructs (∅internal = 1.0 cm; ∅external = 1.5 cm) were implanted subcutaneously on NOD/SCID mice (Jackson Laboratory) maintained in pathogen-free conditions with irradiated chow. For the ex-vivo activation studies in the day of the experiment, NB4 cells were suspended in serum-free medium with (i) no NPs, (ii) with empty NPs (10 µg ml$^{-1}$) or RA$^+$NPs (10 µg ml$^{-1}$) for 4 h. At the end, cells were washed by centrifugation (1,300 r.p.m., 5 min), and the ones treated with RA$^+$NPs were either activated or not with a blue laser (405 nm, 80 mW) for 5 min. NB4 cells (5 × 10$^6$ cells per PDMS construct) were injected subcutaneously in the centre of the PDMS construct embedded in Matrigel (200 µl, BD Biosciences). Five days after injection of the cells, animals were killed by cervical dislocation and cells within the cylindrical construct were collected and characterized by flow cytometry. For the in vivo activation studies in the day of the experiment, NB4 cells were suspended in serum-free medium with (i) no NPs, (ii) with RA$^+$NPs (10 µg ml$^{-1}$) for 4 h. At the end, cells were washed by centrifugation (1,300 r.p.m., 5 min), and 5 × 10$^6$ NB4 cells per PDMS construct were injected subcutaneously in the centre of the PDMS construct embedded in Matrigel (200 µl). One day after injection, experimental groups were either activated or not with a blue optical fibre (405 nm, 80 mW) for 5 min. Three days after injection of the cells, animals were killed by cervical dislocation and cells within the cylindrical construct were collected and characterized by flow cytometry.

**Homing of leukaemia cells to leukaemic niches in vivo.** NOD.CB17-Prkdcscid/J (NOD/SCID) female mice, 6–8-week-old, received 1.5 Gy of total body irradiation from a 137Cs source and was treated with mouse anti-CD122 monoclonal antibody (200 µg) by intraperitoneal injection. Human GFP-labelled THP-1 cells were employed as the target cells and human THP-1 cells without GFP label were employed as delivery cells. On day 0, the target cells (10,000,000 cells per mouse) were injected into the NOD/SCID mice by intravenous injection through the tail vein. On day 1, the delivery cells (10,000,000 cells per mouse; cells were suspended in 200 µl of PBS) loaded with RA$^+$NPs were injected intravenously through the tail vein. The loading of cells was performed as follows. Cells were incubated for 4 h with TRITC-labelled RA$^+$NPs (20 µg ml$^{-1}$) in RPMI medium, washed with PBS to remove loosely bounded NPs, resuspended in RPMI medium with 10% FBS for 18 h and then injected in the animals that had received the target cells. After 24 (n = 3) or 48 h (n = 3) of the delivery cell injection, the mice were killed, the craniums collected and examined with a confocal microscope.

**Bone marrow modulation in vivo.** NOD.CB17-Prkdcscid/J (NOD/SCID) female mice (n = 12) aged 6–8 weeks were employed in this experiment. Before cell injection, mice received 1.5 Gy of total body irradiation from a 137Cs source and were also treated with 200 µg of mouse anti-CD122 monoclonal antibody (NS122) by intraperitoneal injection. Human THP-1 cells were incubated with 20 µg ml$^{-1}$ of RA$^+$NPs in RPMI medium for 4 h, followed by extensive wash with PBS to remove non-internalized NPs. The cells were then resuspended in RPMI medium with 10% FBS and left in the culture incubator overnight. On the following day, the cells loaded with RA$^+$NPs were collected and 1 × 10$^7$ cells per mouse in 200 µl PBS were injected into the NOD/SCID mice intravenously, through the tail vein. After 6 days, the mice were randomly divided into two groups: one of the groups was blue laser irradiated (n = 6) and the other group was not irradiated (n = 6). In the blue laser irradiated group, each mouse was anaesthetized with 3.5% chloral hydrate in PBS and the craniums were exposed to a blue laser (405 nm, 80 mW) for 5 min. The mice in the non-irradiated group received the same treatment without blue laser activation. After 48 or 72 h, the mice were killed and the long bones/craniums were collected. The long bones including the femurs, tibias and pelvis were crushed in PBS containing 1% bovine serum albumin (BSA, sigma) and 2 mM EDTA (Invitrogen). The samples were then filtered with 70 µm cell strainer and treated with ACK solution to lyse red blood cells. The resulting cells were stained using PE-conjugated anti-human CD45 antibody (eBioscience). The results were measured by flow cytometry and the data were analysed with FlowJo software.

The mouse craniums were collected for the ex vivo staining examination followed by the traditional protocol. Briefly, the mouse craniums were cut into four pieces along the sutures producing one frontal, two parietal and one occipital bone. Only parietal bones were used in this experiment. Bone pieces were fixed in 4% paraformadehyde for 30 min, washed twice with PBS, blocked with 2% BSA-0.01%

Triton X-100 in PBS (BSA buffer) for 1 h, and incubated with primary antibodies (diluted 1:100 in BSA buffer) overnight at 4 °C. After washing with BSA buffer for 2 h, bones were incubated with secondary antibodies (diluted 1:200 in BSA buffer) for 2 h, washed with BSA buffer for 2 h and nuclei were stained using DAPI. The primary antibodies included mouse anti-human CD45 antibody (BD) and rabbit anti- human CD11b antibody (Abcam). The secondary antibodies included goat anti-mouse 555 (Invitrogen) and goat anti-rabbit 488 antibodies. The bone pieces were finally examined with a confocal microscope (Nikon Eclipse Ti) under FITC/TRITC/DIC signal channel. Images were analysed in imageJ. Analyse Particles was used after thresholding to quantify positive signals and measure the total area occupied by these particles.

**CRABP2 expression on mouse cranium.** NOD.CB17-Prkdcscid/J (NOD/SCID) female mice aged 6–8 weeks were employed in this experiment. The mouse craniums were collected from the last experiment. In this experiment, we also collected the mouse craniums without any treatment as the control. The *ex vivo* staining examination was followed by the traditional protocol. Briefly, the mouse craniums were cut into four pieces along the sutures producing one frontal, two parietal and one occipital bone. Only parietal bones were used in this experiment. Bone pieces were fixed in 4% paraformadehyde for 30 min, washed twice with PBS, blocked with 2% BSA-0.01% Triton X-100 in PBS (BSA buffer) for 1 h and incubated with primary antibodies (anti-CRABP2 antibody, ab74265, diluted 1:100 in BSA buffer) overnight at 4 °C. After washing with BSA buffer, bones were incubated with secondary antibodies (goat anti-rabbit 488 antibodies, diluted 1:200 in BSA buffer) for 2 h and then washed with BSA buffer. Nuclei were stained using DAPI. The bone pieces were finally examined with a confocal microscope (Nikon Eclipse Ti) under DAPI/FITC/DIC signal channel. Images were analysed in imageJ. Analyse Particles was used after thresholding to quantify positive signals and measure the total area occupied by these particles. Comparison between CD11b, CD45 and CRABP2 positive areas was used as a proxy of the release of RA by NPs and paracrine effect in the parietal bones of irradiated and non-irradiated mice.

All data generated or analysed during this study are included in this published article (and its Supplementary Information files).

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

## Acknowledgements

We acknowledge the use of the Laboratório de Bio-imagem de Alta Resolução of the Faculty of Medicine of University of Coimbra. The authors would like to thank the financial support of FEDER through the programme COMPETE and by Portuguese funds through FCT (PTDC/CTM-NAN/120552/2010 and UID/NEU/04539/2013 to R.P.N.), FCT (SFRH/BD/62419/2009 to C.B.; SFRH/BD/90964/2012 to E.Q.; POCI-01-0145-FEDER-016390:CANCEL STEM to L.F.), EC (ERC project n° 307384, 'Nanotrigger' to L.F.), MINECO (SAF2012-32810, SAF2015-64420-R and Red de Excelencia Con-solider OncoBIO SAF2014-57791-REDC to I.S.-G.) and Bloodwise and CRUK pro-gramme grants to T.E.

## Author contributions

C.B. contributed conceptually to the design of the nanoparticle system and together with E.Q. carried out cell and animal studies, analysed data and wrote the manuscript; Y.C. and D.H. contributed with animal studies and analysed data; A.M.-L., M.B.G.C. and I.S.-G. contributed for animal studies; S.P. carried out cell characterization analyses; R.G. and T.E. contributed for the cell studies; R.P.d.N. and L.F. contributed conceptually, made measurements, analysed data and wrote the manuscript. All the authors have read and approved the manuscript.

**Additional information**

**Competing interests:** The authors declare no competing financial interests.

