## [Peer Review File · Nature Communications]

Reviewers' comments:

Reviewer #1 (Remarks to the Author):

In this manuscript, Boto et al. reported light-sensitive polymeric retinoic acid (RA)-containing nanoparticles (NPs) for modulating the differentiation of leukemia cells. The NPs are composed of light-sensitive photochrome-modified poly(ethyleneimine) (PEI) and dextran sulfate (DS). The in vitro experimental results show that the RA loaded NPs can accumulate in the cytoplasm of leukemia cells for several days and release their RA payloads within a few minutes upon the UV/blue light irradiation, leading to the decrease in the clonogenicity and increase in the differentiation of four types of leukemia cell lines (AML, U937, NB4 and THP-1). The in vivo experimental results show that leukemia cells transfected with RA loaded NPs can engraft into bone marrow in the proximity of other leukemia cells, differentiate upon light irradiation and release paracrine factors to modulate nearby cells. The manuscript is well organized and most experiments are well executed, which support the claims and conclusion of this work. However, the authors should consider following critiques to improve the quality of their manuscript prior to publication.

1. The authors indicated the light-sensitive NPs are stable and well-dispersed in different culture media (Supplementary Fig.3). However, there is severe aggregation for the NPs on the SEM image (Supplementary Fig.2). More explanation of this aggregation phenomenon should be provided.

2. Long term In vitro RA release without UV/blue light irradiation is strongly recommended. From the in vitro results in Fig. 2C1 and in vivo results in Fig. 4B4, without UV/blue light irradiation, there is no differentiation of leukemia cells transfected with RA loaded NPs over three days. This result may suggest that there is zero release of RA from the NPs in the leukemia cells within 3 days, which is unlikely for drug loaded NPs. The authors need to demonstrate drug release and provide an explanation for the observed results.

3. Free drugs like RA are small molecules and their uptake depends on the free diffusion from extracellular to intracellular environment. Therefore, the cellular uptake of free drugs should be faster and higher than that of drug loaded NPs, for which their cellular uptake involves different pathways and endosomal escape. However, the data in Fig. 2D4 shows that the cellular uptake of free RA is much lower than that of RA loaded NPs. More explanation should be provided to discuss this difference.

4. The authors are encouraged to additionally discuss the disadvantages and limitations of their NP platform. For example, it is generally difficult to consistently achieve light-sensitive drug control release in vivo due to the weak penetration of UV light into body tissue.

5. Minor comments:

1) The chemical structure of 4,5-dimethoxy-2-nitrobenzyl chloroformate (DMNC) in Supplementary Fig.1 B is incorrect.

2) The chemical structure of PEI-DMNC in Supplementary Fig.1 B means 100% substitution of the amine groups on PEI. This is inconsistent with experimental results that only 25% of amine groups on PEI are substituted.

3) Two Supplementary Fig.14s are in the manuscript.

Reviewer #2 (Remarks to the Author):

The manuscript entitled "Prolonged intracellular accumulation of light-inducible nanoparticles in leukemia cells allows their remote activation" by Boto and colleagues shows a light-activated drug delivery system. The authors use retinoic acid and acute promyelocytic leukemia cell lines to evaluate the system.

The model is interesting, however the data shown here is not demonstrating elimination of cancer propagating cells as indicated in the text.

The authors need to clearly compare and contrast in their figures the difference between soluble-

RA (+/- light), NP-RA (+/- light) and NP-control (+/- light) for all their experiments. Some of the data is found in some figures, it will help the reader to be able to compare and contrast data. In addition the authors need to compare and contrast APL cell lines (expected to respond to RA) and AML cell lines (with out mutations or translocations that will render them sensitive to RA).

The authors use primary AML cells, the authors did not discuss whether the cells are normal karyotype or PML-RARA.

The authors do not show the toxicity to normal cells or other tissues.

Subcutaneous implants of NB4 cells are not an accurate representation of leukemic tumors.

The authors state: "Leukemia-propagating cells that resist therapy reside in microenvironmental niches in the bone marrow that are difficult to reach by conventional therapy" and to address this they engrafted leukemia cells transfected with RA+NP. This is not showing that the strategy can get to the BM niche. The assay is showing that they can activate the NP when the cells containing them are in the mouse. All in vivo assays were done using transfected cells, which does not demonstrate the feasibility of the particles to reach the leukemia cell in their niche specifically without going into other cells.

Finally, the authors make a big emphasis on leukemia propagating cells and the authors did not performed serial re-plating or serial transplantation.

REVIEWER 1

1. The authors indicated the light-sensitive NPs are stable and well-dispersed in different culture media (Supplementary Fig.3). However, there is severe aggregation for the NPs on the SEM image (Supplementary Fig.2). More explanation of this aggregation phenomenon should be provided.

The authors have performed additional image analysis to further confirm the non-aggregation of the nanoparticles. The aggregation process observed in the SEM analyses was likely an artifact during the drying process of the nanoparticles. Now, in the revised version of the manuscript (Supplementary Fig. 2A), the authors present a representative TEM image of the nanoparticle formulation.

2. Long term In vitro RA release without UV/blue light irradiation is strongly recommended. From the in vitro results in Fig. 2C1 and in vivo results in Fig. 4B4, without UV/blue light irradiation, there is no differentiation of leukemia cells transfected with RA loaded NPs over three days. This result may suggest that there is zero release of RA from the NPs in the leukemia cells within 3 days, which is unlikely for drug loaded NPs. The authors need to demonstrate drug release and provide an explanation for the observed results.

The authors would like to clarify that the *in vitro* results in Fig. 2C1 indicate the release of RA from NPs without UV/blue light irradiation since there is an increase in the differentiation of leukemic cells (U937) from day 1 to day 3. Similar results have been obtained in Fig. 2D2 for leukemic cell line NB4. The *in vivo* results in Fig. 4B4 suggest that the cells that reach the bone marrow are the ones that are less affected by the passive diffusion of RA (RA released without light activation) within the cells. Therefore, the cells that engraft are likely the ones that have less leaky RA⁺NPs and thus are undifferentiated leukemic cells (CD11b^{negative} phenotype). In the revised version of the

manuscript, the authors have included the long-term *in vitro* RA release results from RA⁺NPs without UV/blue light irradiation (**Supplementary Fig. 3**). The authors show that the release of RA is very slow overtime. The authors have also included the following note in the manuscript (page 12): “Overall, our results show that leukemia cells transfected with RA⁺NPs could (i) engraft into the bone marrow and localize at nearby resident leukemia cells, (ii) differentiate after blue laser activation and (iii) modulate the activity/phenotype of the resident leukemia cells. It is possible that the cells that reach the bone marrow are the ones that are less affected by the passive diffusion of RA (i.e., RA release without light activation) within the cells. Therefore, the cells that engraft are likely the ones that have less leaky RA⁺NPs and thus are undifferentiated leukemic cells (CD11b^{negative} phenotype)”.

3. Free drugs like RA are small molecules and their uptake depends on the free diffusion from extracellular to intracellular environment. Therefore, the cellular uptake of free drugs should be faster and higher than that of drug loaded NPs, for which their cellular uptake involves different pathways and endosomal escape. However, the data in Fig. 2D4 shows that the cellular uptake of free RA is much lower than that of RA loaded NPs. More explanation should be provided to discuss this difference.

The authors have included a note in the manuscript to explain the differences in the uptake of free RA and RA loaded NPs. Now, in page 8, the authors stress: “The RA is a hydrophobic small molecule with a water solubility of 63 mg/L (Szuts *et al.*, Archives Biochemistry Biophysics). *In vivo*, RA is transported by proteins called cellular retinoic acid-binding proteins (CRABP) both in extracellular and intracellular compartments (Huang *et al.* Chemical Reviews 2014, 114, 233). These proteins bind RA and transport through the cytoplasm and nucleus up to the nuclear receptor. The expression of CRABP is relatively low in undifferentiated leukemic cells (Douer *et al.*, PNAS 1982, 69, 277) and thus the transport of RA from the outside to the inside of the cell might be limited. In contrast, the transport of RA⁺NPs occurs by endocytosis which seems to facilitate the intracellular accumulation of RA”.

4. The authors are encouraged to additionally discuss the disadvantages and limitations of their NP platform. For example, it is generally difficult to consistently achieve light-sensitive drug control release in vivo due to the weak penetration of UV light into body tissue.

The authors have included an initial statement about the weak penetration of UV light (page 12): “The present research reports a blue laser/UV-inducible NP. The *in vivo* applications of this light trigger are limited, as the optical properties of biological tissue cause attenuation and scattering of light rays to reduce their penetration depth. The recent advances in developing NPs activated by near-infrared light[2] may provide a better option for enhancing light penetration in tissues.” Now in the revised version of the manuscript (page 13) the authors have added the following information: “The present research reports a blue laser/UV-inducible NP. The platform was designed for initial proof of concept regarding the modulation of leukemic cell niches in the bone marrow. Recent studies have explored UV/blue laser approaches to trigger *in vivo* the presentation of bioligands with spatial temporal control to regulate cell adhesion, inflammation, and vascularization of biomaterials (Lee *et al.*, Nature Materials 2015, 14, 352) and to detect protease activity at sites of disease (Dudani *et al.*, ACS Nano 2015, 9(12), 11708). However, the *in vivo* applications of this light trigger are limited, as the optical properties of biological tissue cause attenuation and scattering of light rays to reduce their penetration depth. The recent advances in developing NPs activated by near-infrared light [2] may provide a better option for enhancing light penetration in tissues. For example, upconversion nanocrystals that convert near infrared light into visible light (blue light) may be an interesting possibility for the photocleavage of linkers to which

RA is chemically bound (Want et al., Nature 2010, 463, 1061).”

In the revised version of the manuscript, the authors have also performed blue laser attenuation studies through the skin and calvaria bone (**Supplementary Fig. 19**). The attenuation values are in the range of the ones reported previously by Andres Garcia lab in murine skin [3].

5. Minor comments:

1) The chemical structure of 4,5-dimethoxy-2-nitrobenzyl chloroformate (DMNC) in Supplementary Fig.1 B is incorrect.

The reviewer is right and the authors have performed the correction in the chemical structure.

2) The chemical structure of PEI-DMNC in Supplementary Fig.1 B means 100% substitution of the amine groups on PEI. This is inconsistent with experimental results that only 25% of amine groups on PEI are substituted.

The authors have changed the chemical structure to reflect the degree of substitution.

3) Two Supplementary Fig.14s are in the manuscript.

The authors have changed the labeling of Supplementary Fig. 14.

REVIEWER 2

The manuscript entitled "Prolonged intracellular accumulation of light-inducible nanoparticles in leukemia cells allows their remote activation" by Boto and colleagues shows a light-activated drug delivery system. The authors use retinoic acid and acute promyelocytic leukemia cell lines to evaluate the system. The model is interesting, however the data shown here is not demonstrating elimination of cancer propagating cells as indicated in the text.

The authors would like to clarify that the intention of this study was to demonstrate the remote modulation of leukemic cell niches and not to demonstrate the elimination of leukemia cells. In fact, it would be difficult to activate our UV/blue laser NP system in the entire animal. This would need the development of near-infrared NP systems (discussed in the last part of the manuscript) or other remote activation platform. Yet, the reviewer will agree that inducing differentiation inside the calvarium bone marrow niche by remote activation has never been accomplished before. This proves the principle that it is possible to differentiate leukemia cells when these are in the leukemic niche and in this way decrease the leukemic potential in this microenvironment. Now, in page 13 of the revised manuscript, the authors stress: “The aim of the study was to show the feasibility in remotely induce the differentiation of leukemic cells in the bone marrow and not to demonstrate the exhaustion of cancer propagating cells, as this will be difficult using the blue laser activation process.”

The authors need to clearly compare and contrast in their figures the difference between soluble-RA (+/- light), NP-RA (+/- light) and NP-control (+/- light) for all their experiments. Some of the data is found in some figures, it will help the reader to be able to compare and contrast data. In addition the authors need to compare and contrast APL cell lines (expected to respond to RA) and AML cell lines (with out mutations or translocations that will render them sensitive to RA).

The authors have evaluated the effect of RA and RA⁺NPs (+/- light) in both APL (NB4 and U937; **Figs. 2C and 2D**) and AML cells (primary AML cells: **Fig. 2B**; THP-1 cells: **Supplementary Fig. 12B**). The authors have performed additional experiments to include the controls suggested by the reviewer and have included them in Figures **2B.2, 2C.2, 2D.2** and **Supplementary Fig. 12B**.

The authors use primary AML cells, the authors did not discuss whether the cells are normal karyotype or PML-RARA.

The authors have included the missing information the revised version of the manuscript. Now, in the supplementary information of the revised manuscript, the authors have added the following information: “The AML cells were isolated from a 85 years old man patient with AML 34+117+33+13+DR+ 35% blasts and a 70 years old woman with RAEB2/evolving AML 34+117+33+ 12% blasts. Both samples had formal karyotyping/extended FISH panels. Briefly, neither patients were PML-RARA: one was standard risk (normal karyotype with a small trisomy 8 subclone), and the other was high risk with complex karyotype”.

The authors do not show the toxicity to normal cells or other tissues.

The authors have complemented the previous cytotoxicity results in leukemic cells with cytotoxicity assays in non-leukemic cells such as human endothelial cells (HUVECs) and human fibroblasts. The results are now presented in **Supplementary Fig. 6**. The following information was added to the revised version of the manuscript (pages 4-5): “RA⁺NPs had no substantial effect on leukemia cell metabolism, as evaluated by an ATP assay for concentrations up to 20 µg/mL (**Supplementary Fig. 4A**; in case of non-leukemic cells no cytotoxicity was observed for RA⁺NPs up to 10 µg/mL- **Supplementary Fig. 6**).”

Subcutaneous implants of NB4 cells are not an accurate representation of leukemic tumors.

The authors have performed the subcutaneous implants of NB4 cells to show that the RA⁺NPs could be activated *in vivo* by the blue laser irradiation. The leukemic cells loaded with RA⁺NPs and embedded in Matrigel were injected in a PDMS construct implanted subcutaneously in the animal. After 24 h, the animal was irradiated to activate the RA⁺NPs. Cell differentiation was followed by flow cytometry. We selected this subcutaneous model because we can measure the blue laser attenuation through murine skin (now presented as **Supplementary Fig. 19A**) and we can easily remove the cells from the Matrigel plug and evaluate their differentiation. The authors have now clarified this point in the manuscript (page 10): “We investigated whether leukemic cells loaded with RA⁺NPs could be activated *in vivo* after subcutaneous transplantation. We selected a Matrigel plug subcutaneous model because we know what is the attenuation of the blue laser through the murine skin (**Supplementary Fig. 19A**) and we can easily remove the cells from the Matrigel plug and evaluate their differentiation by flow cytometry. Initially we evaluated whether the *in vivo* environment could interfere with the differentiation program of leukemic cells after *ex vivo* activation (...).”

The authors state: "Leukemia-propagating cells that resist therapy reside in microenvironmental niches in the bone marrow that are difficult to reach by conventional therapy" and to address this they engrafted leukemia cells transfected with RA+NP. This is not showing that the strategy can get to the BM niche.

The authors are not sure if they understood correctly the comment of the reviewer. The authors show that leukemia cells carrying RA⁺NPs can get to the BM niche. The strategy that we propose here is that cells can be used as a delivery agent. Our data suggests that we can envisage a therapeutic option that uses the natural capabilities of bone marrow cells to find the bone marrow niche. We show that once they reach the niche the therapeutic cargo can be deployed (by light trigger) and released by these cells via paracrine routes in the niche microenvironment.

The assay is showing that they can activate the NP when the cells containing them are in the mouse. All in vivo assays were done using transfected cells, which does not demonstrate the feasibility of the particles to reach the leukemia cell in their niche specifically without going into other cells.

The use of nanoparticles to target bone marrow has limitations (off-target effects). The objective of the current study was to use leukemic cells as delivery agents. We suggest a shift of paradigm where cells are the delivery agent as they find their own way to the niche. The light-activation trigger for the release of the therapeutic agent decreases the potential for off-target effects. Now in page 13 of the revised version of the manuscript, the authors stress: “We have used leukemic cells as a programmable delivery system that recognizes the leukemic niches and modulate them after light activation. The concept here is different from previous studies that have used therapeutic NPs coated with aptamers to direct them to the bone marrow. Although the results achieved were interesting, the side effects of the systemic administration of the NPs containing a powerful anti-leukemic agent remain to be determined. In the present study we have used cancer biology to fight cancer. We have used leukemia cells loaded with a clinical-approved AML differentiating agent to home at leukemic niches in the bone marrow. In a clinical setting, the leukemia cells could be isolated from the patient that would receive the treatment. Although not explored in the current study, normal hematopoietic stem cells may act as well as a nanoparticle carrier to target the leukemic niches”.

Finally, the authors make a big emphasis on leukemia propagating cells and the authors did not performed serial re-plating or serial transplantation.

The authors agree with the reviewer and they have removed the word “propagating” from the manuscript. Indeed, the authors have not demonstrated the “propagation” properties of the THP-1 cell line in the bone marrow, which was outside the scope of the current study. As mentioned before, the aim of the study was to demonstrate the remote modulation of leukemic cell niches.

REFERENCES

1. Duan CW, Shi J, Chen J, Wang B, Yu YH, Qin X, et al. Leukemia propagating cells rebuild an evolving niche in response to therapy. *Cancer Cell* 2014;25(6):778-93.
2. Shanmugam V, Selvakumar S, Yeh CS. Near-infrared light-responsive nanomaterials in cancer therapeutics. *Chem Soc Rev* 2014;43(17):6254-87.
3. Lee TT, Garcia JR, Paez JI, Singh A, Phelps EA, Weis S, et al. Light-triggered in vivo activation of adhesive peptides regulates cell adhesion, inflammation and vascularization of biomaterials. *Nat Mater* 2015;14(3):352-60.

Reviewers' comments:

Reviewer #1 (Remarks to the Author):

Boto et al. reported light-sensitive polymeric retinoic acid (RA)-containing nanoparticles (NPs) for modulating the differentiation of leukemia cells. Upon UV/blue light irradiation, this drug delivery platform can release its payload within a few minutes upon the, leading to the decrease in the clonogenicity, increase in the differentiation of leukemia cells, and release of paracrine factors to modulate nearby cells in vivo. In the revised manuscript, the authors have addressed most of the concerns raised by the reviewers and have conducted additional experiments including examination of long term RA release and evaluation of the cytotoxicity of the drug loaded NPs against normal tissue cells (e.g., human endothelial cells and human fibroblasts). While these additional results support the significance of this work, the following issues still need to be addressed before publication in Nature Communications.

1. It is not representative to show only one NP in the TEM image (Supplementary Figure 2A). A representative TEM showing the dispersity of the NPs need to be provided.
2. The substitution degree (DS) of PEI-DMNC in Supplementary Figure 1B is 50%, which is inconsistent with experimental results of 10% of amine groups on PEI being substituted (Supplementary Figure 1A). The authors need to revise the structure of PEI-DMNC (e.g., using "R" to represent DMNC or amine group in the structure).

Reviewer #2 (Remarks to the Author):

The authors have address all the concerns. They have greatly improved the manuscript by clearly describing that the delivery of the nanoparticles is by the leukemia cells. They also propose potential future applications and remove all comments/speculation that were not supported by the data.

REVIEWER 1

1. It is not representative to show only one NP in the TEM image (Supplementary Figure 2A). A representative TEM showing the dispersity of the NPs need to be provided.

The authors have now performed additional TEM image analyses to illustrate the dispersity of the NPs.

2. The substitution degree (DS) of PEI-DMNC in Supplementary Figure 1B is 50%, which is inconsistent with experimental results of 10% of amine groups on PEI being substituted (Supplementary Figure 1A). The authors need to revise the structure of PEI-DMNC (e.g., using “R” to represent DMNC or amine group in the structure).

The authors have revised the chemical structure according to the indications of the reviewer.